# Risk Factors of Proximal Screw Breakage of Locking Plate (ZPLP^®^) after MIPO for Distal Femur Fractures -Analysis of Patients with Plate Removal after Bony Union-

**DOI:** 10.3390/jcm12196345

**Published:** 2023-10-03

**Authors:** Jehyun Yoo, Daekyung Kwak, Joongil Kim, Seungcheol Kwon, Junhyuk Kwon, Jihyo Hwang

**Affiliations:** 1Department of Orthopaedic Surgery, Hallym University Sacred Heart Hospital, Anyang-si 14068, Republic of Korea; oships@hallym.or.kr (J.Y.); limitkd@hallym.or.kr (D.K.); 2Department of Orthopedic Surgery, Kangnam Sacred Heart Hospital, Hallym University College of Medicine, 1 Singil-ro, Yeongdeungpo-gu, Seoul 07441, Republic of Korea; jungil@hanmail.net (J.K.); youthinl@naver.com (S.K.); junhyuck367@hallym.or.kr (J.K.)

**Keywords:** screw breakage, locking plate, bony union, distal femur

## Abstract

Background: Locking a compression plate is a more favorable surgical technique than intramedullary nailing in the treatment of distal femur fractures. This study analyzed the risk factors of proximal screw breakage retrospectively, which was confirmed in the patients with plate removal after bony union. Methods: A total of 140 patients who were fixed by MIPO using ZPLP from 2009 to 2019 were identified. A total of 42 patients met the inclusion criteria and were included. The screw breakage group (12 patients) and the non-breakage group (30 patients) were compared. Results: Approximately 12 (28.6%) of 42 plate-removal patients showed proximal screw breakage. The breakage of proximal screws developed at the junction of the screw head and neck. The number of broken proximal screws averaged 1.4 (1~4). The breakage of the proximal screw even after the bony union is more frequent in older patients (*p* = 0.023), the dominant side (*p* = 0.025), the use of the cortical screw as the proximal uppermost screw (*p* = 0.039), and the higher plate-screw density (*p* = 0.048). Conclusions: Advanced age, dominant side, use of the cortical screw as the uppermost screw, and higher plate-screw density were related to proximal screw breakage. When the plate is removed after bony union or delayed union is shown in these situations, the possibility of proximal screw breakage should be kept in mind.

## 1. Introduction

The incidence of femoral fractures varies in the literature and is reported to be approximately 3–6% [1,2,3,4,5,6]. Other studies that have been undertaken suggest a world incidence of 9.0–22.8/1000/year [2]. Distal femur fractures account for 4–6% of osteoporosis-related fractures of the femur in the elderly population [7]. In the Republic of Korea, the incidence of fractures is not well established. The increase in knee joint arthroplasty surgeries has led to a higher incidence of distal femur periprosthetic fractures. The rates of periprosthetic fractures are 0.3 to 2.5% after primary total knee arthroplasty (TKA) and 1.6 to 38% after revision TKA [8]. The distal femur has been known as a surgically challenging area due to the significant displacement after fracture, making it difficult to achieve anatomical reduction through open reduction methods [9,10]. Due to the transition to the shaft and distal parts of the femur, intramedullary nails are biomechanically stable; however, correcting angular deformities is difficult [11]. On the other hand, plating allows for angular deformity correction but has the disadvantage of being inferior to intramedullary nailing in terms of biomechanical stability [12,13,14,15,16]. Therefore, choosing the appropriate implant is difficult, and in cases where non-union occurs, reoperation can be quite challenging. Among various surgical treatments, the use of a locking head screw plate with the ability to correct angular deformities without the need for a knee incision has become the gold standard treatment for distal femur fractures, as it offers advantages over retrograde intramedullary nailing. However, compared to the nail, the locking head screw plate has its disadvantages. In some cases, the screw may break during the bone healing process or after the union is achieved [11]. This can occur as a positive process for achieving union or as a negative process due to the lack of mechanical stability leading to breakage, ultimately resulting in non-union. The minimally invasive technique of locking a compression plate is a well-recognized method in the treatment of distal femur fractures, and the principle behind this technique has been studied [17]. Proper technique and surgical results have been reported; however, little research has been conducted on this topic up to now. A fixation failure study has been reported in case series [11,18] but any studies have not revealed the risk factors for screw breakage after the union of the fracture site. In this study, we aimed to identify the risk factors for locking screw breakage, particularly after complete union is achieved during metal removal. The authors analyzed the number and pattern of broken screws to identify the appropriate plate length, screw density, and type that can prevent breakage and thus prevent potential complications.

## 2. Patients and Methods

Between December 2009 and January 2021, 140 patients (52 males and 88 females) aged 15 to 93 years underwent minimally invasive plate osteosynthesis (MIPO) using the Zimmer^®^ Periarticular Locking Plate (ZPLP^®^) for distal femoral fracture. Three experienced surgeons in two different university-affiliated hospitals performed these operations. This was a retrospective study based on medical documents and radiologic images. 

Distal femoral fractures with AO classification types IIIA, IIIB, and IIIC were included in the list. The exclusion criteria were patients with fixation failure and nonunion, patients fixed by dual plating, and those who did not undergo plate removal surgery.

Of 140 patients, 98 were excluded from this study, and 42 met the inclusion criteria. Finally, 12 patients in whom proximal screw breakage was confirmed intraoperatively when the plate was removed were included in the screw breakage group, and 30 patients were included in the non-breakage group (Figure 1). The demographics and clinical outcomes of the two groups were compared in this study. This study was approved by the Institutional Review Board of Gangnam Sacred Heart Hospital, Hallym University (HKS 2021-4-002-002). 

### 2.1. Surgical Technique

Skeletal traction below the knee was applied until surgery for the prevention of muscle contracture and fracture site shortening. Patients were positioned supine on the radiolucent operating table under appropriate anesthesia. The lower leg was draped from the iliac crest to the foot for intraoperative assessment of the length, rotation, and angulation. The standard lateral parapatellar approach was used with the knee in 30-degree flexion supported by a rolled operation sheet. An incision was made over the lateral aspect of the distal femur, and the vastus lateralis muscle was retracted to expose the fracture site. With fluoroscopy, reduction of fracture was confirmed with coronal and sagittal views, and the Zimmer^®^ Periarticular Locking Plate (ZPLP^®^) was inserted submuscularly and preliminarily fixed with temporary K-wires or drill bits before the insertion of real screws. 

Before definitive screw fixation of the plate, coronal alignment and sagittal alignment were checked again, and definite plate fixation with screws was performed. If the plate contour is not acceptable, which means the gap between the plate and bone is separated, a temporary cortical screw is used to reduce the gap between the plate and bone. The temporary cortical screw can remain or be removed. 

Rehabilitation started on the second postoperative day, changing from a long leg splint to a Velcro knee brace and continuous passive motion of the hip and knee joints. After discharge, patients were encouraged to perform straight leg-raising exercises. Non-weight bearing was recommended in patients with intraarticular fractures until callus bridging was visualized on plain radiographs; otherwise, tolerable weight bearing was taught. Coronal and sagittal plane angulations were assessed on anteroposterior and lateral full-length femurs, and knee x-ray films were captured immediately after surgery and at every outpatient follow-up visit. Generally, the patients were discharged from the hospital after 2 weeks of operation, which can be followed up every month until union is gained. 

### 2.2. Evaluation

Preoperative factors include age, gender, fracture classification, fracture configurations, femoral bowing, injury mechanism, time to injury, BMD, BMI, comorbidities, and ASA score. Fracture classification was classified by the AO classification, which defines 33-A as extraarticular, 33-B as partial articular, and 33-C as a complete articular fracture. Fracture configuration was classified into oblique, spiral, and transverse patterns. The ASA Physical Status Classification System was defined as ASA I as a normal healthy patient, ASA II as a patient with mild systemic disease, ASA III as a patient with severe systemic disease, and ASA IV as a patient with severe systemic disease that is a constant threat to life. 

Intraoperative factors include operation time, plate span width, screw density, screw fixation, or wiring of the fracture site. 

Each group’s differences in mean values of continuous data were compared using the Student t-test, and the association of categorical data were examined using the Pearson chi-squared test. The significance level was set at *p* < 0.05, and the statistical software used was SPSS 24.0 (IBM Corp., Armonk, NY, USA).

## 3. Results

There was a huge incidence of screw breakages. Approximately 12 (28.6%) of 42 plate-removal patients showed screw breakages. All screws were located proximal to fracture sites. The breakage of proximal screws developed at the junction of the screw head and neck. The number of broken proximal screws averaged 1.4 (1~4).

An 83-year-old female patient showed multiple screw breakages at the junction of the screw head and neck after the removal of the plate (Figure 2). An 81-year-old male patient who was reduced by LCP for periprosthetic fractures remained with two broken screws after the removal of the plate (Figure 3). 

Risk factors were analyzed, distinguishing between preoperative and postoperative factors.

### 3.1. Preoperative Risk Factor

In the screw-breakage group, the mean age was 67.4, the ratio of sex (male:female) was 2 to 12, BMI (kg/m^2^) was 26.0, BMD (T-score) was −2.37, site (Rt:Lt) was 9 to 3, and the average ASA classification was 2.1. 7 patients had an underlying disease; 1 patient was diagnosed with obesity, and 1 patient was diagnosed with diabetes mellitus. In the non-screw-breakage group, the mean age was 55.5, the ratio of sex (male:female) was 10 to 20, BMI (kg/m^2^) was 24.3, BMD (T-score) was −1.87, site (Rt:Lt) was 11 to 19, and the average ASA classification was 1.9. 17 patients had an underlying disease; 2 patients were diagnosed with obesity, and 7 patients were diagnosed with diabetes mellitus (Table 1).

### 3.2. Intraoperative Risk Factor

In the screw-breakage group, the mean operation time was 121 min, the length of the plate was 273 mm, 13.2 mm (hole), the plate span width was 3.53, the empty holes were 8.1, the wiring of patients was 4, the total number of filled screws was 11.3, the proximal was 4.25, the fracture site was 0.25, and the distal was 6.83. The plate screw density was a total of 0.53, proximal was 0.48, fracture site was 0.16, and distal was 0.96, and there were 6 fixation crossings over the fracture line. In the non-screw-breakage group, the mean operation time was 130 min, the length of the plate was 279 mm, 13.2 mm (hole), the plate span width was 4.93, the empty holes were 8.6, the wiring of patients was 5, the total number of filled screws was 10.7, the proximal was 3.96, the fracture site was 0.40, and the distal was 6.53. The plate screw density was a total of 0.47, proximal was 0.44, fracture site was 0.14, and distal was 0.91, and there were 10 fixation crossings over the fracture line (Table 2). 

Non-modifiable preoperative risk factors were age (*p* = 0.023) and site of operation (*p* = 0.025) (Rt:Lt). Modifiable intraoperative surgical factors were plate screw density (*p* = 0.048) and screw configurations (*p* = 0.039). 

## 4. Discussion

This study investigates the factors contributing to screw breakage in minimally invasive plate osteosynthesis for distal femur fractures and explores ways to prevent this complication. The breakage rate of 28.6% of cases was a highly remarkable result. The results show that older age, the dominant side, the use of a cortical screw as the proximal uppermost screw, and higher plate-screw density are associated with a higher frequency of screw breakage. 

Old age, that is, osteoporotic bone, shows a risk factor of screw breakage, which is a well-known risk factor and can be easily overlooked by the surgeon. This is a non-modifiable risk factor; therefore, strict implant sections and surgical techniques are needed. The main pitfall in plate osteosynthesis of old-age osteoporotic bone is the risk of creating a too-stiff construct that stresses the concentration of screws. Quality of reduction, screw type, screw configuration, the length and position of the plate, and the working length of the construction are important to elderly patients [19]. Elderly patients tend to be less cooperative about weight bearing, which can affect their stress concentration. 

There was no study about the dominant and non-dominant risk factors. The authors predict that screw breakage occurs more frequently in the dominant leg simply because it is used more often, leading to the accumulation of stress. 

Previous studies have examined the effects of plate-screw density, plate length, and number of screws on MIPO using LCP for distal femur fractures. The appropriate plate-screw density is recommended to be 0.4–0.5, and a minimum of 12 screw holes are required [13,20,21]. Stoffel et al. [22] reported that three or more screws in each bone do not significantly increase axial stiffness and that four or more screws do not increase torsional rigidity. The working length of the plate and the plate span width also affect axial stiffness and torsional rigidity. If the working length of the plate is too short, fixation failure may occur due to high shear forces at the fracture site. The plate span width is appropriate at 2–3 times for comminuted fractures and 8–10 times for simple fractures [20,23,24,25,26].

What we previously knew was that inserting screws near a fracture site is associated with an increased risk of internal fixation failure [27]. Using locking screws when fixing the most proximal screw in the metaphysis can cause stress risers and lead to Young’s modulus fractures due to their larger diameter and higher concentration of stress [28]. To prevent internal fixation failure and valgus deformity, the distal screws should be parallel to the joint surface, and the plate should be parallel to the outer cortical bone of the femoral shaft. Some studies suggest that using cortical screws to apply compression and then using locking screws to fix them in place during the insertion of screws into the proximal femur can be useful [29].

Harvin et al. [30] stated that all proximal locking screws are the most significant predictors of non-union. Among 43 fractures treated with all proximal locking screws, non-union occurred in 21 (48.8%), while among 52 fractures treated with hybrid proximal screws, non-union occurred in 13 (25.0%). Bottlang et al. [28] reported that using a cortical screw instead of a locking screw in the proximal holes of a locking plate reduces stress concentration at the end of the plate and increases resistance to bending. Due to these issues, far-cortical locking (ZPLP) has been developed to make locking plates more flexible, and it has been shown to significantly increase bone callus formation and torsional resistance compared to standard locking constructs [31].

However, using cortical screws in the proximal part of the hybrid LCP increases the risk of screw breakage, as post-surgical stress amplification points occur at the transition from high area moment of inertia (AMI) to lower area moment of inertia [32]. This means that the change in AMI occurs at the conventional cortical screw hole, as the screw is inserted in a lag fashion and is not considered part of the implant [33]. Therefore, it is important to choose the appropriate implant and follow the recommended rules to minimize complications that may occur after surgery.

Implant removal is not a routine procedure; when the surgeon considers the removal of the implant, screw breakage, and screw head stripping should be considered. These problems are so troublesome during the operation. If the surgeon prevents this problem, implant stiffness and material are also important. Moldevan et al. emphasized torch control during the insertion of screws, which are especially made of weak materials such as titanium alloy [34].

Our study has limitations, such as a small number of cases and not being a comparative study with conventional plate fixation. Additionally, it is a retrospective study. Furthermore, only patients who achieved bone union were included, and cases of non-union or delayed union were excluded. Finally, the current study results showed that the plate length and number of screw holes did not affect the outcomes.

## 5. Conclusions

The use of the MIPO technique with an LCP for the treatment of distal femoral fractures in the elderly has recently been reported to yield satisfactory results. The 28.6% incidence of screw breakage is a striking finding in this study.

Even though LCP showed relatively good results, stress concentration on the screw and plate junction still occurred even after the bony union. Even though the exact risk factors were not figured out in this study due to the lack of cases, the authors suggest that the use of a locking screw at the proximal uppermost is recommended for the prevention of screw breakage after bony union. The stress on the locking screw head and neck junction is higher than that on cortical screws. If the whole screws were locking screws, this stress on the screw head and neck junction could be decreasing. Too high a density of screws also risks factors of screw breakage after gaining bony union. When such a surgical method is chosen, caution is required, as screw breakage occurs when the plate is removed after gaining bone union, despite the significant surgical benefits. Therefore, the surgeon should approach the treatment with caution, considering the possibility of breakage during the removal of the screw when using a cortical screw in elderly patients who underwent surgery on their dominant leg.

## Figures and Tables

**Figure 1 jcm-12-06345-f001:**
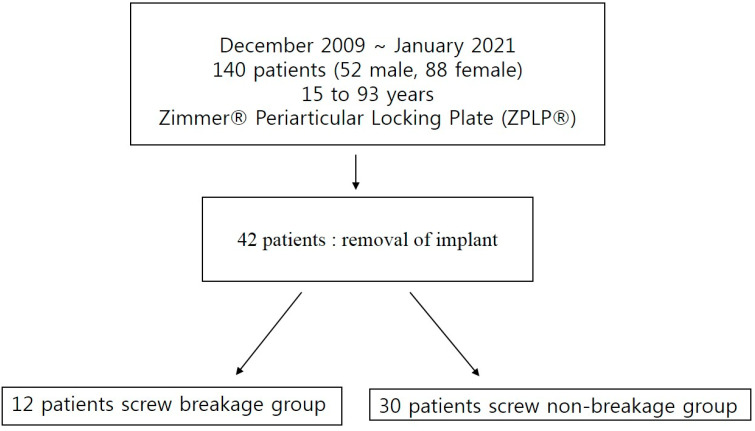
The flow chart of this study.

**Figure 2 jcm-12-06345-f002:**
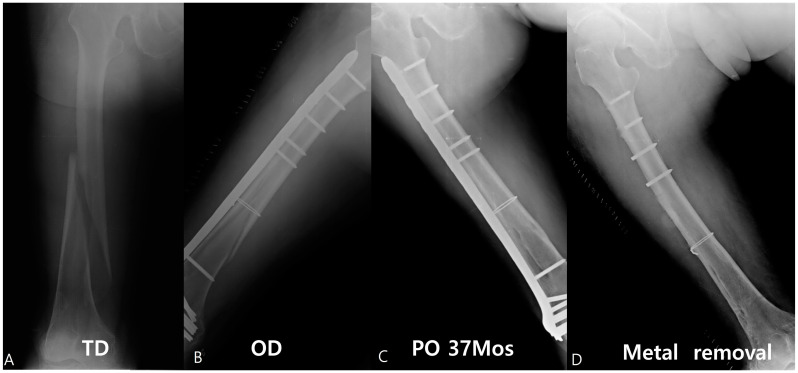
An approximately 83-year-old female patient was reduced by the Zimmer^®^ Periarticular Locking Plate (ZPLP^®^). The fracture pattern was a long spiral with low energy (**A**). A total of 18 holes, one plate, and one wire were used for reduction (**B**). This patient was followed up until 37 months (**C**), and the plate was removed, which showed four broken screws (**D**).

**Figure 3 jcm-12-06345-f003:**
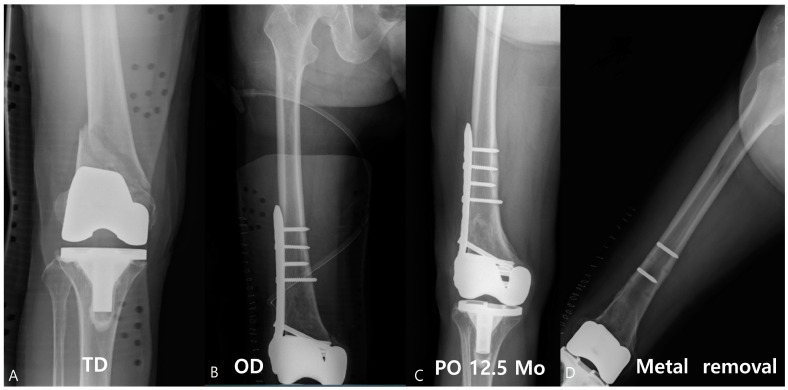
Approximately 81-year-old male patient visited with a periprosthetic fracture after total knee arthroplasty (**A**); 10 holes of LCP were used for the reduction (**B**). After union, the plate was removed at 12.5 months, and the two screws were broken (**C**,**D**).

**Table 1 jcm-12-06345-t001:** Statistical analysis on preoperative risk factors.

Risk Factor	Screw Breakage Group (N = 12)	Non-Breakage Group (N = 30)	*p* Value
Age	67.4	55.5	0.023
Sex (M:F)	2:12	10:20	0.18
BMI (kg/m^2^)	26.0	24.3	0.20
DEXA (T-score)	−2.37	−1.87	0.18
Rt:Lt	9:3	11:19	0.025
ASA	2.1	1.9	0.32
Underlying disease	7	17	0.92
Obesity	1	2	0.85
DM	1	7	0.26
Smoking	0	1	0.52
Dialysis	0	0	N/A
Postop period (Mo)	24.8	21.3	0.60
AO classification(A:B:C)	11:0:1	19:1:10	0.18
Fx. configuration1. Comminuted 2. Spiral 3. Oblique 4. Transverse	5:3:2:2	9:8:6:7	0.89
Periprosthetic fracture	2	4	0.78
Time from injury to surgery (day)	3	2.3	0.32

**Table 2 jcm-12-06345-t002:** Statistical analysis on intraoperative risk factors.

Risk Factor		Screw Breakage Group (N = 12)	Non-Breakage Group (N = 30)	*p* Value
Operation Time (min)		121	130	0.59
Length of plate (mm)		273	279	0.72
Length of plate (holes)		13.2	13.2	1.00
Plate span width		3.53	4.93	0.17
Empty holes		8.1	8.6	0.62
Wiring		4	5	0.23
Filled screws	Total	11.3	10.7	0.16
	Proximal	4.25	3.96	0.12
	Fracture site	0.25	0.40	0.45
	Distal	6.83	6,53	0.25
Plate screw density	Total	0.53	0.47	0.048
	Proximal	0.48	0.44	0.47
	Fracture site	0.16	0.14	0.82
	Distal	0.96	0.91	0.16
Fixation crossing fracture line		6	10	0.31
Screw at the proximal uppermost hole				
1: cortical		9	12	
2: locking unicortical		0	11	
3: locking bicortical		3	7	0.039

## Data Availability

Not Applicable.

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
