# Peer review of "Risk Factors of Proximal Screw Breakage of Locking Plate (ZPLP®) after MIPO for Distal Femur Fractures -Analysis of Patients with Plate Removal after Bony Union-"

_jcm, 2023, doi:10.3390/jcm12196345_

Round 1

Reviewer 1 Report

An interesting publication and methodologically well-conducted study.

Comments below:

1) It is worth working on the introduction. General information on the epidemiology of distal femoral fractures is lacking. There is also no information on what risk factors for screw breakage have been described so far.

2) The methodology is well prepared.

3) The results section is well written but monotonous. It is worth presenting the results in a way that is more accessible to the reader

4) In the discussion, little space was devoted to the influence of age and the dominant leg on the screw breakage

5) All citations should be written in English.

Author Response

1) It is worth working on the introduction. General information on the epidemiology of distal femoral fractures is lacking. There is also no information on what risk factors for screw breakage have been described so far.

We added more information and citation

2) The methodology is well prepared.

3) The results section is well written but monotonous. It is worth presenting the results in a way that is more accessible to the reader

We changed

4) In the discussion, little space was devoted to the influence of age and the dominant leg on the screw breakage

We comment about the age and dominant leg

5) All citations should be written in English.

We changed to English citation

Reviewer 2 Report

The research aim is to analyze the risk factors associated with breakage of proximal skews during osteosynthesis material removal in distal femur fractures.

The paper does not respect the template of the journal.

The abstract is structured appropriately.

The introduction transposes the research into the topic and formulates the objective of the study at the end.

In the methodology section, there is no information on the statistical analysis performed. A flow chart of the study sample should be provided for a better overview. In the Evaluation subsection the first sentance has no point as the idea is repetead in the next paragraphs.

The results of the study are not clarly described.

The discussions is too short and needs to be extended and related to more scientific papers. For example, information about the importance of torque control of the screws in relation to osteosynthesis failure could be addressed in relation to Flaviu Moldovan, Tiberiu Bățagă. Torque Control during Bone Insertion of Cortical Screws. https://doi.org/10.1016/j.promfg.2020.03.070.

The conclusions should be related more to the actual findings of the paper.

The references should be edited according to the journals criteria and should be also extended as suggested above

The tables should be edited in APA style. As a general rule of reporting the p values: if p value is greater than 0.05 should be reported with two decimal values, if p value is between 0.001 and 0.05 should be reported with three decimal places; I suggest to make this correction throughout the paper.

Author Response

The research aim is to analyze the risk factors associated with breakage of proximal skews during osteosynthesis material removal in distal femur fractures.

The paper does not respect the template of the journal.

We respected the template of the journal according to the instructions for authors

The abstract is structured appropriately.

We shortened the words within 200

The introduction transposes the research into the topic and formulates the objective of the study at the end.

We changed

In the methodology section, there is no information on the statistical analysis performed. A flow chart of the study sample should be provided for a better overview. In the Evaluation subsection the first sentance has no point as the idea is repetead in the next paragraphs.

We added information of statistical ananlysis

We made flow chart

1st sentence is removed

The results of the study are not clarly described.

We made more clearly

The discussions is too short and needs to be extended and related to more scientific papers. For example, information about the importance of torque control of the screws in relation to osteosynthesis failure could be addressed in relation to Flaviu Moldovan, Tiberiu Bățagă. Torque Control during Bone Insertion of Cortical Screws. https://doi.org/10.1016/j.promfg.2020.03.070.

We comment about the torch control and cited this article in the discussion part

The conclusions should be related more to the actual findings of the paper.

We changed conclusions

The references should be edited according to the journals criteria and should be also extended as suggested above

We added and changed references

The tables should be edited in APA style. As a general rule of reporting the p values: if p value is greater than 0.05 should be reported with two decimal values, if p value is between 0.001 and 0.05 should be reported with three decimal places; I suggest to make this correction throughout the paper.

We changed according to your comment

Round 2

Reviewer 2 Report

The authors have adressed all my concerns.